# The Sorting Nexin Genes *ChSNX4* and *ChSNX41* Are Required for Reproductive Development, Stress Adaption and Virulence in *Cochliobolus heterostrophus*

**DOI:** 10.3390/jof8080855

**Published:** 2022-08-15

**Authors:** Huilin Yu, Wantong Jia, Zhongxiang Li, Chaofeng Gao, Hongyu Pan, Xianghui Zhang

**Affiliations:** College of Plant Science, Jilin University, Changchun 130062, China

**Keywords:** *Cochliobolus heterostrophus*, sorting nexin, reproductive development, virulence

## Abstract

Sorting nexins are a conserved protein family involved in many cellular processes in fungi, and the function of sorting nexin Snx4 (Atg24) and Snx41 (Atg20) in *Cochliobolus heterostrophus* was not clear. The Δ*Chsnx4* and Δ*Chsnx41* mutants were generated by a PCR-based marker method to determine the roles of Snx4 and Snx41 in reproductive development, stress adaption, and virulence in *C. heterostrophus*. Compared with the wild-type strain, the Δ*Chsnx4* and Δ*Chsnx41* mutants exhibited obvious changes in vegetative growth and in morphology of conidia. In addition, the conidiation, appressorium formation, and virulence of snx4 and snx41 mutants were dramatically reduced. Moreover, Δ*Chsnx4* and Δ*Chsnx41* mutants were more sensitive to oxidative stress (menadione and H_2_O_2_), cell wall integrity stress (Congo red and calcofluor white), fungicides, and isothiocyanates. All the phenotypes mentioned above were restored in complemented strains. In addition, ChSnx4 and ChSnx41 were proven to interact with each other through yeast two-hybrid. Taken together, these findings indicated that *ChSNX4* and *ChSNX41* were important for fungal growth, asexual development, stress adaption, and virulence in *C. heterostrophus*.

## 1. Introduction

The endosomal system consists of a complicated network of several compartments and functions designed to sort internalized cargo to the appropriated subcellular destinations. In sorting pathways, early endosomes (EEs) are important components and are responsible for cargo proteins’ trafficking and sorting [1,2]. In this process, a series of sorting machineries are involved. Sorting nexins (SNXs) are a large group of proteins with phosphoinositide-binding PX (phox homology) domain that have been involved in endosomal sorting regulation [3,4,5]. In addition to the PX domain, some sorting nexins also contains a C-terminal BAR (Bin/Amphiphysin/Rvs) domain, which are members of the SNX-BAR subfamily [6]. 

A number of sorting nexins play important roles in the recognition and transportation of cargo proteins. In budding yeast, 7 of 15 PX domain proteins were found to have fundamental roles in protein sorting [6]. Vps5 and Vps17, the best-known sorting nexins in yeast, form a an SNX-BAR dimer that cooperates with the cargo selective subcomplex to form the pentameric retromer complex [7]. In *Saccharomyces cerevisiae*, Snx41, Snx4, and Atg20 were involved in the trafficking of Snc1 and Can1 [8,9]. In addition, Snx4 (Cvt13 or Atg24) and Snx42 (Cvt20 or Atg20) in *S. cerevisiae* are necessary for Ape1 transportation in the cytoplasm-to-vacuole targeting (Cvt) pathway and are involved in autophagy [10]. Moreover, suppression of SNX4 perturbs transport between the endocytic recycling compartments and causes lysosomal degradation of the transferrin receptor TfnR [11]. Furthermore, these homologs have also played essential roles in phytopathogenic fungi. MoSnx41, an ortholog of yeast Snx41 in *Magnaporthe oryzae*, is required for conidiation, virulence, pexophagy and protein sorting [12]. In addition, MoSnx41 played an important role in oxidative stress adaption during infection in *M. oryzae* [13]. Moreover, deletion of MoAtg24, an orthologue of yeast *SNX4*, leads to defects in conidiation and mitophagy in *M. oryzae* [14]. In *Fusarium graminearum*, Snx41 interacted with Snx4, and the Snx41–Snx4 complex involved in FgSnc1 recycling, vegetative development, and virulence [15]. Deletion of *FgATG20/FgSNX42* resulted in defective asexual development, pathogenicity, Cvt, macroautophagy, and pexophagy [16]. In *Podospora anserine*, *PaATG24* (*PaSNX4*) also played an essential role in growth, fertility, and lifespan [17].

The functions of conserved regulatory proteins often show species-specific variations, so it was of interest to study the two nexin genes in a necrotrophic plant pathogen. Southern corn leaf blight caused by *C. heterostrophus* is a destructive foliar disease in the world. To dissect the roles of ChSnx4 and ChSnx41 in *C. heterostrophus*, the Δ*Chsnx4* and Δ*Chsnx41* mutants were constructed, and we revealed that ChSnx4 and ChSnx41 are required for mycelial growth, conidiation, stress adaption, and virulence.

## 2. Materials and Methods

### 2.1. Fungal Strains and Plant Materials

*C. heterostrophus* wild-type strain C4 (*Tox1+, MAT1-2*, ATCC 48331) was used for gene deletion and cultivated on complete medium with xylose (CMX) at 25 °C in the dark [18,19]. Maize inbred line “B73” was used for virulence test.

### 2.2. Identification of ChSNX4 and ChSNX41

Amino acid sequences of Snx4 and Snx41 proteins in *S. cerevisiae* (NP_012498.1 and NP_010170.1, respectively) were used as blast queries against the genome of *C. heterostrophus* strain C4. The top candidates in *C. heterostrophus* were used in a reciprocal blast against the Joint Genome Institute Mycocosm (JGI) database and only the blast hits with the correct domain were considered as orthologs and used for further research. A phylogenetic tree was constructed based on amino acid sequences to confirm the orthologs in *C. heterostrophus*. Snx4 and Snx41 amino acid sequences with the top hits from *S. cerevisiae*, *Neurospora crassa*, *Magnaporthe oryzae*, *Sclerotinia sclerotiorum*, *Aspergillus nidulans*, *Fusarium oxysporum*, *Setosphaeria turcica*, *Botrytis cinerea* and *Fusarium graminearum* were included in the phylogenetic tree. Proteins were aligned using the MEGA6 program′s maximum likelihood method.

### 2.3. Construction of Snx4 and Snx41 Mutants and Complementation Strains, and PCR Verification

The split-marker method was used to delete the *SNX4* and *SNX41* genes [20]. In brief, the whole promoter-*HYG* gene-terminator fragment was amplified from plasmid pUCATPH [21] using primers M13F and M13R. The primer pair FP1/RP1 was used to amplify the 5’ flanking region of the *SNX4/SNX41* gene, FP2/RP2 was used to amplify the 3′ flanking region of *SNX4/SNX41* gene, and 5′ and 3′ flanking fragments were fused to the *HYG* gene, respectively. The fused fragments were used to transform *C. heterostrophus*, and transformants were selected on CMX medium without salts containing hygromycin B [19]. Diagnostic PCR was used to verify the transformants, which included primer pair Chsnx4-F/Chsnx4-R (Chatg20-F/Chatg20-R), Chsnx4-UF (Chatg20-UF)/NLC37, and Chsnx4-DR (Chatg20-DR)/NLC38 (Appendix A). For complementation, three fragments were amplified, including the WT *SNX4/SNX41* genes with 5′ and 3′ flanking sequences, a fragment further 3′ of the *SNX4/SNX41* gene 3′ flanking sequence, and the *NPT*Ⅱ gene from plasmid pⅡ99. These three fragments were co-transformed into Δ*snx4/snx41* mutants, geneticin (400 μg/mL) was used as a selectable marker, and *gen^R^* transformants were grow on CMX without salts (CMXNS) for 3–5 d and confirmed by PCR.

### 2.4. Phenotypic Analysis

To compare the colony morphology of mutants to WT, a mycelial plug of each strain was placed on CMX medium and incubated at 25 °C in the dark. After 4 d, the colony diameters of mutants and WT were measured, and colony morphologies were photographed. In addition, to measure the conidiation and germination rate of conidia, 7-day-old cultures on CMX were used to harvest conidia, and mycelial debris was removed by filtration. The germination rate of conidia was determined on a glass slide, and 50 conidia were counted for each sample. In addition, the morphology and appressorium formation of the conidia was observed. Three replicates were set up for each experiment, and each experiment was repeated three times.

### 2.5. Virulence Test

For the virulence test, 10 μL conidial suspension (50 conidia/μL) of WT, mutants, and complemented strains were used to inoculate 3-week-old “B73” maize detached leaves. Inoculated leaves were transferred to a petri dish and kept for 24 h, then were moved out and kept at 25 °C under 16 h of light/8 h of dark. Lesion areas were evaluated 4 d after inoculation. Three leaves were used for each replicate, and the virulence test was repeated three times.

### 2.6. Yeast Two-Hybrid (Y2H) Assay

The interaction between Snx4–Snx41 was verified by Y2H. The full-length cDNA of *SNX4* was amplified and cloned into the vector pGBKT7 to construct the bait plasmid. The cDNA sequence of SNX41 was amplified and cloned into the vector pGADT7 to construct the prey plasmid. After being confirmed by sequencing, both prey and bait plasmids were co-transformed into yeast strain AH109. Transformants were verified on SD-Trp-Leu-His-Ade medium. The interactions between pGBKT7-Lam and pGADT7-T and between pGBKT7-53 and pGADT7-T were used as negative control and positive control, respectively.

### 2.7. Response to Stress and ROS Detection

To investigate the sensitivities of the Δ*snx4* and Δ*snx41* mutants to different stresses, all strains were inoculated on potato dextrose agar medium (PDA), minimum medium (MM), minimum medium without nitrogen (MM-N), and CM supplemented with different agents, which included Congo red (CR), calcoflour white (CFW), menadione, and H_2_O_2_. In addition, the Δ*snx4* and Δ*snx41* mutants, WT, and complemented strains were also inoculated onto CMX amended with fungicides (Tebuconazole, Azoxystrobin and Boscalid) and isothiocyanates (ITCs). After 7 days, the colony diameter of each strain was measured. To observe the ROS production, the mycelia of WT and mutants were stained with 0.5 mg/mL 3,3′-diaminobenzidine (DAB) for 12 h.

### 2.8. Expression Level Analysis

To investigate the expression levels of *ChSNX4* and *ChSNX41* in different developmental stages (hypha, conidia, germination stage, and infection stage), samples were collected and ground in liquid nitrogen. The RNA was extracted with a TransZol Up Plus RNA kit (TransGen Biotech Co., Ltd., Beijing, China) and purified with an Ambion TU RBO DNA-free kit (Applied Bio Systems, Darmstadt, Germany). Then, RNA was reverse transcribed into cDNAs (TransScript All-in-One First-Strand cDNA Synthesis SuperMix for qPCR, TransGen Biotech Co., Ltd., Beijing, China). Quantitative Real-Time PCR was used to detect the expression levels of *ChSNX4* and *ChSNX41*, and the *actin* gene (Primers were shown in Appendix A) was used as an internal reference. qRT-PCR was repeated three times.

### 2.9. Statistical Analysis

For statistical analysis, GraphPad prism program′s *t*-tests and multiple *t*-tests were used for significant differences analysis. All data shown are the mean ± standard error of the median (SEM).

## 3. Results

### 3.1. Identification of SNX4 and SNX41 in C. heterostrophus

Snx4 (ChSnx4, JGI protein ID131530) and Snx41 (ChSnx41, JGI protein ID170417) were identified in *C. heterostrophus*. The ChSnx4 protein contains 506 amino acids, and ChSnx41 contains 645 amino acids. The predicted polypeptide encoded by *ChSNX4* and *ChSNX41* both showed the characteristic phosphoinositide-binding PX domain (N-terminal) and BAR domains (C-terminal) (Figure 1B). Phylogenetic analysis indicated that Snx4 and Snx41 were conserved in phytopathogenic fungi and *S. cerevisiae* (Figure 1A). Phylogenetic analysis indicated that ChSnx4 and ChSnx41 were mostly closest to StSnx4 and StSnx41, respectively.

Through real-time PCR, we found both *ChSNX4* and *ChSNX41* are more highly expressed in the conidia stage than in the hyphae stage, especially for *ChSNX4*, for which the expression level was increased three times. In the germination stage, with the extension of time, the expression levels of *ChSNX4* and *ChSNX41* also increased. In the infection stage, the expression levels of *ChSNX4* and *ChSNX41* at 24 h were highest (Figure 1C). These results indicated that *ChSNX4* and *ChSNX41* may be involved in conidiation and infectious hypha growth in *C. heterostrophus*. 

In addition, Snx4 was reported to interact with Snx41 in *F. graminearum* [15]. In this study, ChSnx4–ChSnx41 interaction was confirmed by yeast two-hybrid assay (Appendix A).

### 3.2. ChSNX4 and ChSNX41 Were Required for Proper Vegetative Growth

After 7 days of incubation on CMX, Δ*Chsnx4* and Δ*Chsnx41* exhibited an obviously slower growth rate than the wild-type and complemented strains. In addition, Δ*Chsnx4* and Δ*Chsnx41* mutants produced uneven edge and dark green aerial mycelia (Figure 2A,B). The same results were also observed on PDA, MM, and MM-N (Figure 2C,D), which suggested that *ChSNX4* and *ChSNX41* were crucial for proper vegetative growth.

### 3.3. ChSNX4 and ChSNX41 Disruption Caused Defects in Asexual Reproduction

Quantitative analysis indicated that the Δ*Chsnx4* mutant was reduced by 68.1% in conidiation compared with the WT, and that the Δ*Chsnx41* mutant was reduced by 75.4% (Figure 3B). In addition, in comparison with the WT, the conidial lengths of the Δ*Chsnx4* and Δ*Chsnx41* mutants were much shorter, but the conidial width of the Δ*Chsnx4* and Δ*Chsnx41* mutants showed no obvious difference (Figure 3A,C,D). Moreover, the conidia germination rates and germ tube length were affected in mutants (Figure 4A–C). The conidia of Δ*Chsnx4* and Δ*Chsnx41* mutants showed obvious reduction in appressorium formation; only 52.7% and 54.7% of germinated conidia of Δ*Chsnx4* and Δ*Chsnx41* mutants, respectively, formed appressoria compared with 87.3% for WT on 4 h (Figure 4D). 

### 3.4. ChSNX4 and ChSNX41 Are Necessary for Stress Adaption

To determine the role of *ChSNX4* and *ChSNX41* in cell wall integrity, the Δ*Chsnx4* and Δ*Chsnx41* mutants, WT, and complemented strains were inoculated on CMX with CR and CFW. The colony growth of the Δ*Chsnx4* and Δ*Chsnx41* mutants was significantly inhibited on CMX amended with 300 μg/mL CR and 20 μg/mL CFW (Figure 5A,B); this suggested that *ChSNX4* and *ChSNX41* may play a vital role in cell wall integrity. In addition, compared with WT, the Δ*Chsnx4* and Δ*Chsnx41* mutants were more sensitive to Menadione and H_2_O_2_, especially when the concentration of H_2_O_2_ was increased to 10 mM (Figure 5A,B). Generally, the sensitivity of strains to H_2_O_2_ was related to intracellular ROS production, once the intracellular ROS production was increased, the strains always exhibited more sensitivity to H_2_O_2_. DAB can react with H_2_O_2_ to generate deep-brown polymer; the Δ*Chsnx4* and Δ*Chsnx41* mutants were more heavily stained by DAB than WT and complemented strains (Figure 5C). These results indicated that loss of *ChSNX4* and *ChSNX41* resulted in more intracellular ROS production, and ChSnx4 and ChSnx41 are essential for oxidative stress. 

Moreover, compared with WT and complemented strains, Δ*Chsnx4* and Δ*Chsnx41* mutants exhibited more sensitivity to Tebuconazole and Azoxystrobin, and Δ*Chsnx4* was significantly inhibited on 0.25 and 1 ppm Boscalid (Figure 6A,B). In our previous study (data not published), through transcriptomic analysis, we found that *ChSNX4* was significantly upregulated when treated with isothiocyanate (ITC), one kind of secondary metabolite with antimicrobial effects produced by Cruciferae (Figure 6C). So, the sensitivity of Δ*Chsnx4* to 4-(methylthio)-butyl isothiocyanates (M-ITC) and phenylethyl isothiocyanates (P-ITC) were determined in this study. The results showed that Δ*Chsnx4* mutant was more sensitive to M-ITC and P-ITC (Figure 6D,E). This was consistent with the result in transcriptomic analysis and suggested that *ChSNX4* may be involved in antifungal agent adaption.

### 3.5. ChSnx4 and ChSnx41 Are Necessary for Virulence

To investigate the functions of Snx4 and Snx41 in virulence, the conidia droplets of the WT strain, Δ*Chsnx4*, and Δ*Chsnx41* mutants, and the complemented strains were inoculated on susceptible *Zea mays* “B73”. Typical lesions were caused in the WT and complemented strains, but no lesions were caused in mutants; chlorosis was observed on the detached leaves inoculated with Δ*Chsnx4*, and no symptoms were found on the detached leaves inoculated with Δ*Chsnx41* (Figure 7).

## 4. Discussion

So far, only the orthologs of Snx4 and Snx41 were well studied in *F*. *graminearum* and *M*. *oryzae*, but the functions of conserved regulatory proteins often show species-specific variations, so it was of interest to study the two nexin genes in a necrotrophic plant pathogen *C. heterostrophus*. In the current study, the roles of *ChSNX4* and *ChSNX41* were investigated in *C. heterostrophus*, the causative agent of Southern corn leaf blight. Here, we found that both ChSnx4 and ChSnx41 played essential roles in mycelia growth, asexual development, stress adaption, and virulence in *C. heterostrophus*. In addition, ChSnx4 interacted with ChSnx41 strongly. However, ChSnx4 and ChSnx41 were not necessary for non-selective macroautophagy (data not shown). Sorting nexins are a conserved protein family involved in vesicle transport, membrane trafficking, and protein sorting, so loss of ChSnx4 and ChSnx41 affect a number of core cell biology processes. These pleiotropic phenotypes proved that ChSnx4 and ChSnx41 played an important role in growth and development in *C. heterostrophus*, and this has been verified in *M. oryzae* and *F*. *graminearum*.

In *M. oryzae*, MoAtg24, a sorting nexin related to yeast Snx4, was identified and played an important role during asexual development [14]. In addition, *MGG_12832*, an ortholog of yeast *SNX41* and *ATG20/SNX42*, was also characterized in *M. oryzae* [13]; this means that the *M. oryzae* genome only has one gene, which is homologous to yeast *ScATG20/SNX42* and *ScSNX41*. However, two different homologous genes (FGSG_06950 and FGSG_16745) of *ScATG20/SNX42* and *ScSNX41* were identified in *F. graminearum*. FGSG_16745 was identified as *FgSNX41* [15], and FGSG_06950 was identified as *FgSNX42* [16]. Meanwhile, only one gene (XM_014223294.1) is homologous to yeast *ScATG20/SNX42* and *ScSNX41* in *C. heterostrophus*. Phylogenetic analysis indicated that ChSnx41 was mostly close to StSnx41, and ChSnx4 was mostly close to StSnx4, and both ChSnx41 and ChSnx4 were far away from ScSnx41/Snx4. These results indicated that Snx4 and Snx41/42 are conserved in phytopathogenic fungi but have not been functionally identified in most phytopathogenic fungi.

In this study, the virulence of Δ*Chsnx4* and Δ*Chsnx41* mutants was reduced significantly compared with WT and complemented strains. In *M. oryzae*, Snx41 was found necessary for conidia formation and pathogenesis [13]. Subsequently, He et al. reported that loss of MoSnx4 disrupts mitophagy and consequently leads to highly reduced conidiation, but the pathogenicity of Δ*Mosnx4* was not mentioned [14]. The sorting nexins FgSnx4 and FgSnx41 interact with each other and assemble into a heterodimer through BAR domains [15]. In addition, deletion of either FgSnx4 or FgSnx41 causes defects in the mycelial growth and pathogenicity [15]. As mentioned above, *F. graminearum* possesses two different genes homologous to ScSnx42 and ScSnx41; FgSnx42 was also involved in pathogenicity [16]. In human fungal pathogen *Aspergillus fumigatus*, sorting nexins Snx41/Atg20 were involved in growth, cell wall stress response, and virulence [22]. These limited studies clarified the functions of Snx4/Snx41 in phytopathogenic fungi. Moreover, the conidiation decreased dramatically in Δ*Chsnx4* and Δ*Chsnx41* mutants; a similar phenotype was also characterized in *M. oryzae* and *F. graminearum*. In the field, *C. heterostrophus* is mainly dispersed by asexual spores, so asexual development is a very crucial stage for *C. heterostrophus*. Once conidiation is blocked, the disease caused by *C. heterostrophus* will be easier to control. Furthermore, *ChSNX4* and *ChSNX41* are involved in conidia germination and germ tube formation. The conidia of the Δ*Chsnx4* and Δ*Chsnx41* mutants showed obvious reduction in appressorium formation; only 52.7% and 54.7% of germinated conidia of Δ*Chsnx4* and Δ*Chsnx41* mutants formed appressoria in 4 h. *C. heterostrophus* is a typical necrotrophic fungus; even though the appressoria is not essential for infection, it can help *C. heterostrophus* penetrate maize leaves faster [23]. Therefore, we speculated that the reduced conidial germination rate and appressoria formation rate would be the reasons for the reduced virulence of the Δ*Chsnx4* and Δ*Chsnx41* mutants. In addition, the slower hyphal growth rate maybe also contributes to reduced virulence. 

The sorting nexin Snx4 and Snx41 have been proven to be related to several autophagy pathways and endosomal trafficking. Ablation of PaSNX4/ATG24 in *Podospora anserine* leads to mitophagy and pexophagy disruption [17]. In *S. cerevisiae*, Snx4/Atg24 and Snx41/Atg20 are not only required for the Cvt pathway, but are also required for pexophagy and mitophagy [10,24]. In *M. oryzae*, MoAtg24 was dispensable for pexophagy and macro-autophagy, but played an important role in mitophagy [14]. Just as in the previous studies, in our study, ChSnx4 and ChSnx41 were not necessary for non-selective macroautophagy (data not shown). However, the functions of ChSnx4 and ChSnx41 in selective autophagy were not determined in this study. In the future, we will uncover the functions of ChSnx4 and ChSnx41 by cytological observation and Western blot analysis. In addition, the Snx4–Snx41–Snx42 complex in *S. cerevisiae* was necessary for protein sorting and traffic [8]. FgSnx41 and FgSnx4 interact with each other and assemble into a heterodimer which is required for the endosomal recycling of FgSnc1 [15]. Furthermore, in human pathogenic fungi *A. fumigatus*, physical interaction between Snx41/Atg20 and Snx4/Atg24 was identified and compared with the individual gene deletion mutants, and there was no obvious growth difference in double gene-deletion mutants, suggesting that the function of each protein is independent on the other [22]. In this study, the interaction between ChSnx4 and ChSnx41 was also identified by Y2H; this was consistent with the results mentioned above. In future, we will confirm the interaction between ChSnx4 and ChSnx41 with more approaches and screen the putative cargo proteins of ChSnx4–ChSnx41 in *C. heterostrophus*. 

## 5. Conclusions

In the current study, we have characterized how Snx4 and Snx41 impact virulence. Furthermore, these two nexins are involved in conidiation, which *C. heterostrophus* uses to disseminate in the field. These results may be helpful for designing fungicides that target Snx4 or Snx41 for Southern corn leaf blight control.

## Figures and Tables

**Figure 1 jof-08-00855-f001:**
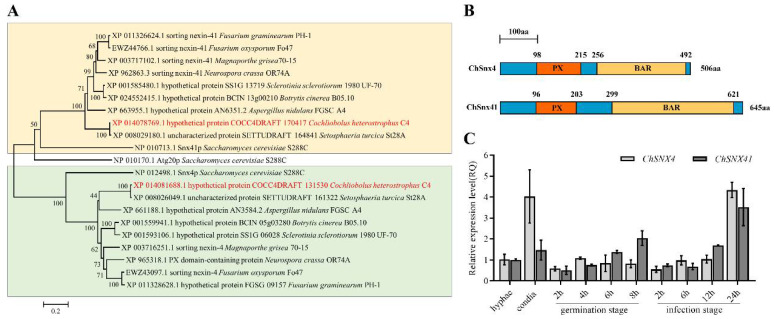
(**A**) Phylogenetic analysis of Snx4 and Snx41 in different organisms. The amino acid sequences of *C. heterostrophus*, *S. cerevisiae*, *A. nidulans*, *S. turcica*, *F. oxysporum*, *F. graminearum*, *M. oryzae*, *B. cinerea*, and *S. sclerotiorum* were used to construct the phylogenetic tree with the MEGA 6 program. (**B**) The information on the PX (phox homology) and BAR (Bin/Amphiphysin/Rvs) domains of ChSnx4 and ChSnx41. (**C**) Relative expression levels of ChSNX4/ChSNX41 during different developmental stages; each experiment had three replicates and was repeated three times.

**Figure 2 jof-08-00855-f002:**
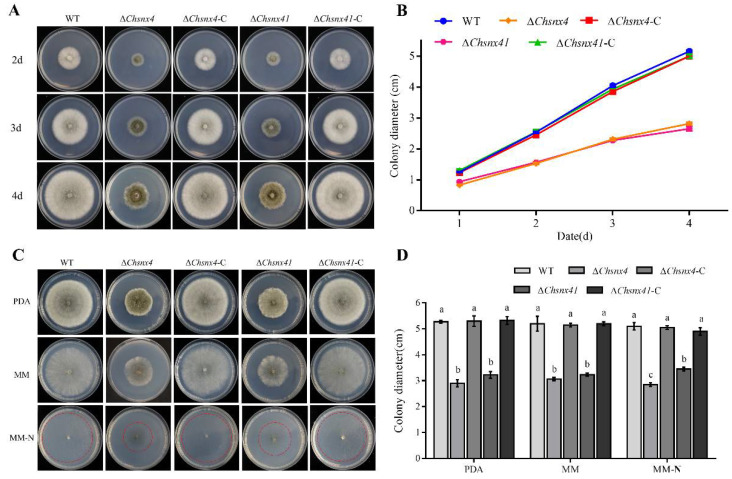
*Cochliobolus heterostrophus* Snx4 and Snx41 are required for colony growth on CMX (CMX: complete medium, glucose was replaced by xylose), PDA, MM, and MM-N. (**A**) Colony morphology of WT, Δ*Chsnx4*, Δ*Chsnx4-C*, Δ*Chsnx41*, and Δ*Chsnx41*-C grown on CMX for 4 days. (**B**) Statistical analysis of the colony diameter of indicated strains on the 4th day. (**C**) Colony morphology of WT, Δ*Chsnx4*, Δ*Chsnx4-C*, Δ*Chsnx41*, and Δ*Chsnx41*-C grown on PDA, MM, and MM-N. (**D**) Statistical analysis of colony diameter on PDA, MM, and MM-N. Each experiment had three replicates and was repeated three times.

**Figure 3 jof-08-00855-f003:**
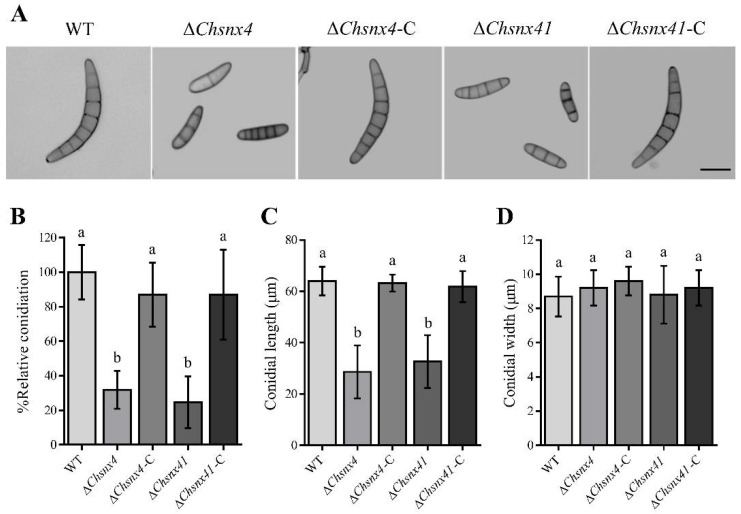
Snx4 and Snx41 are required for conidiation and conidia morphology in *C. heterostrophus*. (**A**) Representative conidial morphology of WT, mutants, and complemented strains. (**B**) Conidiation of indicated strains. X-axis: indicated strains; y-axis: relative conidiation (%). (**C**) Average conidial length of indicated strains. X-axis: indicated strains; y-axis: average conidial length (μm). (**D**) Average conidial width of indicated strains. X-axis: indicated strains; y-axis: average conidial width (μm). Each experiment had three replicates and was repeated three times. (Scale bar 10 μm).

**Figure 4 jof-08-00855-f004:**
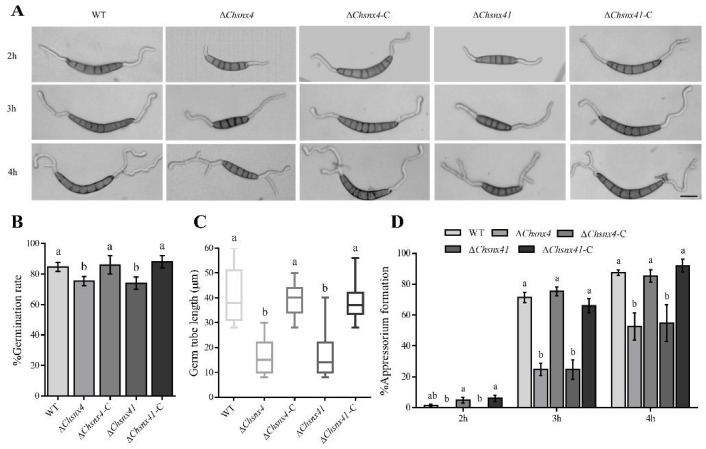
Snx4 and Snx41 are required for conidia germination and germ tube and appressoria formation in *C. heterostrophus*. (**A**) Germ tube and appressoria morphology of WT, mutants, and complemented strains at 2 h, 3 h, and 4 h. (**B**) Conidial germination rate of indicated strains. (**C**) Average germ tube length of indicated strains. (**D**) Appressorium formation rate of the indicated strains at 2 h, 3 h and 4 h. Each experiment had three replicates and was repeated three times. (Scale bar 10 μm).

**Figure 5 jof-08-00855-f005:**
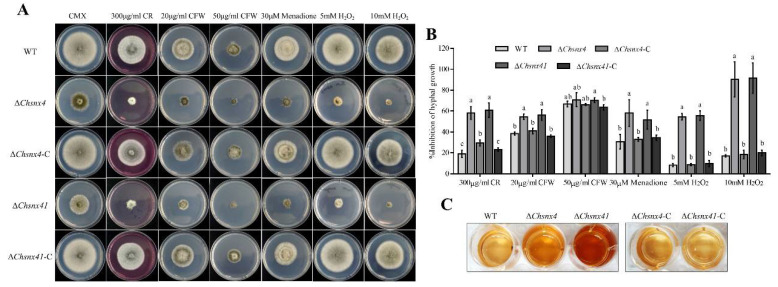
Sensitivity of WT, Δ*Chsnx4*, Δ*Chsnx4-C*, Δ*Chsnx41*, and Δ*Chsnx41*-C to a variety of stress agents. (**A**) Colony morphology of indicated strains grown on CMX supplemented with Congo red (CR), calcofluor white (CFW), Menadione, and H_2_O_2_. (**B**) Statistical analysis of inhibition of hyphal growth of each strain under CR, CFW, Menadione, and H_2_O_2_ stress conditions. (**C**) The mycelia of the indicated strains were stained with DAB. Each experiment had three replicates and was repeated three times.

**Figure 6 jof-08-00855-f006:**
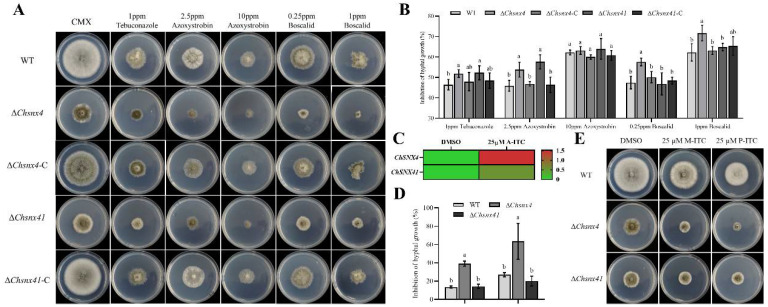
Sensitivity of WT, Δ*Chsnx4*, Δ*Chsnx4-C*, Δ*Chsnx41*, and Δ*Chsnx41*-C to fungicides and isothiocyanates. (**A**) Colony morphology of the indicated strains grown on CMX supplemented with Tebuconazole, Azoxystrobin, and Boscalid. (**B**) Statistical analysis of the inhibition of the hyphal growth of each strain under Tebuconazole, Azoxystrobin, and Boscalid stress conditions. (**C**) The relative expression levels of *ChSNX4* and *ChSNX41* under A-ITC treatment. (**D**) Statistical analysis of the inhibition of the hyphal growth of each strain grown on CMX supplemented with M-ITC and P-ITC. (**E**) Colony morphology of the indicated strains grown on CMX supplemented with M-ITC and P-ITC. Each experiment had three replicates and was repeated three times.

**Figure 7 jof-08-00855-f007:**
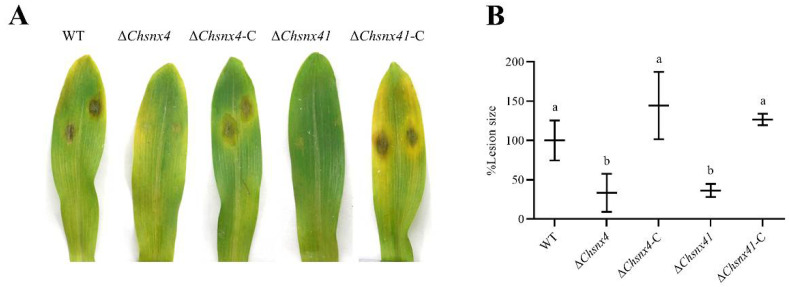
Virulence of WT, mutants, and complemented strains on maize. (**A**) Inoculation with conidia droplets of each strain on detached, susceptible *Zea mays* “B73” leaves. (**B**) Lesion size of the indicated strains on detached maize leaves. Each experiment had three replicates and was repeated three times.

## Data Availability

Not applicable.

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
