# Peer review of "The Sorting Nexin Genes ChSNX4 and ChSNX41 Are Required for Reproductive Development, Stress Adaption and Virulence in Cochliobolus heterostrophus"

_jof, 2022, doi:10.3390/jof8080855_

Round 1
Reviewer 1 Report
Sorting nexins are involved in regulating intracellular membrane traffic, endosomal protein sorting and cytoplasm-to vacuole targeting, autophagy and peroxisome autophagy (pexophagy). There are relatively few previous studies in filamentous fungi in general and plant pathogens in particular (two: Magnaporthe & Fusarium). Here, two nexin genes were characterized in the necrotrophic maize pathogen Cochliobolus heterostrophus, by gene knockout and complementation. It might be somewhat disappointing that cytoplasm-to vacuole targeting and autophagy were not addressed in depth, though the results certainly give the base to continue to address these aspects. The experiments are well-designed and the data support the conclusions.
1. virulence assays - it is not clear if detached leaves were used throughout (Figure 7 notes "intact susceptible leaves"); detached leaves are very sensitive to infection, and the rounded first leaf shown is more susceptible than later emerged mature leaves. That said, the phenotype is very clear, because even under these conditions, the mutants could not infect.
2. The mutants affect a core cell biology process(es) so the phenotypes are quite pleiotropic. This is not necessarily a weakness but could be discussed further.
minor comments:
line 12 and elsewhere - not really "split" marker as the entire construct was amplified to transform with a single linear DNA fragment. This not a problem but might define better (can just say "a PCR-based marker method")
line 26 - consists of
line 30 - "were involved in and played an important role" can shorten to "are involved"
line 35 play important roles in
line 52 resulted in defective asexual development,
line 55 The biological functions ... were not clear. The functions of conserved regulatory proteins often show species-specific variations, so it was of interest to study the two nexin genes in a necrotrophic plant pathogen. [next sentence] - Southern corn leaf blight caused by...
line 85, rather than F/R and U, D - please give the full primer names as listed in Table S1
Complementation was most successful: though it is supposed to be routine, it is not not always; for readers wanting to make use of this in their work - please give concentration of geneticin/G418, media and growth times
line 95 - on the other side - not clear what that means, please rewrite
line 103 complemented
line 103 to inoculate
line 104 - mist chamber - these are detached leaves, correct? so wouldn't a petri dish have been sufficient? if not detached leaves, please clarify (see note above)
line 144 - closest - note that St is taxonomically very close to C. heterostrophus
Figure 1 (and in some other Figures) - font is small and low resolution
line 155 - expression levels - this is found only for ChSNX4
line 158 - may play an essential role [actually, there is no evidence for this, just a correlation made from on the basis of increasing expression]
Figure 2 and elsewhere - letters indicate significance - how was the statistical test done (please note in the methods or legends)
line 204 - may play a vital role
line 207 was related
first sentence of the Discussion - as in the introduction, the rationale for doing the study is not just "not explored in C. heterostrophus" but rather "of interest to study in a necrotrophic plant pathogen" or similar; perhaps combine with the first sentence of the following paragraph
line 292 - important, the sentence says the opposite of what was intended - "even though the appressoria are not essential for infection"
line 319 - In the current study, we have characterized how Snx4 and Snx41 impact virulence. Furthermore these two nexins are involved in conidiation, which C. heterostrophus uses to disseminate in the field.
Figure 6, panel D - labels are missing on the z-axis (growth times?)
Author Response
- virulence assays - it is not clear if detached leaves were used throughout (Figure 7 notes "intact susceptible leaves"); detached leaves are very sensitive to infection, and the rounded first leaf shown is more susceptible than later emerged mature leaves. That said, the phenotype is very clear, because even under these conditions, the mutants could not infect.
Response 1: Thanks. The detached leaves were used throughout, and we have revised in the Figure 7.
- The mutants affect a core cell biology process(es) so the phenotypes are quite pleiotropic. This is not necessarily a weakness but could be discussed further.
Response 2: That’s a very good suggestion. We have discussed in the first paragraph of discussion.
minor comments:
line 12 and elsewhere - not really "split" marker as the entire construct was amplified to transform with a single linear DNA fragment. This not a problem but might define better (can just say "a PCR-based marker method")
Response: Replaced with "a PCR-based marker method".
line 26 - consists of
Response: Revised in manuscript.
line 30 - "were involved in and played an important role" can shorten to "are involved"
Response: Replaced with "are involved" in manuscript.
line 35 play important roles in
Response: Revised in manuscript.
line 52 resulted in defective asexual development,
Response: Revised in manuscript.
line 55 The biological functions ... were not clear. The functions of conserved regulatory proteins often show species-specific variations, so it was of interest to study the two nexin genes in a necrotrophic plant pathogen. [next sentence] - Southern corn leaf blight caused by...
Response: Revised in manuscript.
line 85, rather than F/R and U, D - please give the full primer names as listed in Table S1
Response: The full names were added in MM and Table S1.
Complementation was most successful: though it is supposed to be routine, it is not not always; for readers wanting to make use of this in their work - please give concentration of geneticin/G418, media and growth times
Response: The concentration of geneticin/G418, media and growth times were added in MM.
line 95 - on the other side - not clear what that means, please rewrite
Response: “on the other side” was replaced with “In addition” in manuscript.
line 103 complemented
Response: Revised in manuscript.
line 103 to inoculate
Response: Revised in manuscript.
line 104 - mist chamber - these are detached leaves, correct? so wouldn't a petri dish have been sufficient? if not detached leaves, please clarify (see note above)
Response: Yes, petri dish is sufficient, and revised in manuscript.
line 144 - closest - note that St is taxonomically very close to C. heterostrophus
Response: Revised in manuscript. Yes, St is taxonomically very close to Ch. Thanks.
Figure 1 (and in some other Figures) - font is small and low resolution
Response: Actually, the resolution is high in our original Figures. I will ask the editor for help to make it clearer.
line 155 - expression levels - this is found only for ChSNX4
Response: Yes, it’s only for ChSNX4, so we use “expression level”.
line 158 - may play an essential role [actually, there is no evidence for this, just a correlation made from on the basis of increasing expression]
Response: We have changed to “may involve in”
Figure 2 and elsewhere - letters indicate significance - how was the statistical test done (please note in the methods or legends)
Response: Statistical analysis was added in the MM.
line 204 - may play a vital role
Response: Revised in manuscript.
line 207 was related
Response: Revised in manuscript.
first sentence of the Discussion - as in the introduction, the rationale for doing the study is not just "not explored in C. heterostrophus" but rather "of interest to study in a necrotrophic plant pathogen" or similar; perhaps combine with the first sentence of the following paragraph
Response: Rewrite in manuscript.
line 292 - important, the sentence says the opposite of what was intended - "even though the appressoria are not essential for infection"
Response: Revised in manuscript.
line 319 - In the current study, we have characterized how Snx4 and Snx41 impact virulence. Furthermore these two nexins are involved in conidiation, which C. heterostrophus uses to disseminate in the field.
Response: Revised in manuscript.
Figure 6, panel D - labels are missing on the z-axis (growth times?)
Response: The Figure 6 was replaced with a new one.
Reviewer 2 Report
The objective of this article is to investigate the roles of sorting nexin Snx4 and Snx41 in Cochliobolus heterostrophus. The authors constructed ChSnx4 and ChSnx41 mutants and concluded that ChSNX4 and ChSNX41 were important for reproductive development, stress adaption and virulence. These results will provide some useful information for design fungicides of southern corn leaf blight control.
For this article by itself, it is interesting and well worth posting. The experience is well planned and executed. Discuss the results are correct and clearly written.
There are some issues that best to do before publishing:
All the figure legends need to have information on the number of samples, the number of times the experiment has been replicated and the statistical analysis have carried out.
Line 93-100: font size is different from full text.
Line 134: primers for the actin gene should be provided.
Line 242: how many leaves were used in the virulence test to evaluate the lesion size?
Author Response
All the figure legends need to have information on the number of samples, the number of times the experiment has been replicated and the statistical analysis have carried out.
Response: Replicates and repeats were added in figure legend, and statistical analysis was added in MM. Thanks.
Line 93-100: font size is different from full text.
Response: Revised in manuscript.
Line 134: primers for the actin gene should be provided.
Response: Primers of actin gene were added in Table S1.
Line 242: how many leaves were used in the virulence test to evaluate the lesion size?
Response: Three leaves were used in each replicate.